# Pandemic velocity: Forecasting COVID-19 in the US with a machine learning & Bayesian time series compartmental model

**Gregory L. Watson** [1]*, **Di Xiong** [1], **Lu Zhang** [1], **Joseph A. Zoller** [1], **John Shamshoian** [1], **Phillip Sundin** [1], **Teresa Bufford** [1], **Anne W. Rimoin** [2], **Marc A. Suchard** [1,3], **Christina M. Ramirez** [1]

**1** Department of Biostatistics, Fielding School of Public Health, University of California, Los Angeles, California, United States of America, **2** Department of Epidemiology, Fielding School of Public Health, University of California, Los Angeles, California, United States of America, **3** Departments of Computational Medicine and Human Genetics, David Geffen School of Medicine, University of California, Los Angeles, California, United States of America

\* gwatson@ucla.edu

**Data Availability Statement:** The data underlying the results presented in the study are available

## Abstract

Predictions of COVID-19 case growth and mortality are critical to the decisions of political leaders, businesses, and individuals grappling with the pandemic. This predictive task is challenging due to the novelty of the virus, limited data, and dynamic political and societal responses. We embed a Bayesian time series model and a random forest algorithm within an epidemiological compartmental model for empirically grounded COVID-19 predictions. The Bayesian case model fits a location-specific curve to the velocity (first derivative) of the log transformed cumulative case count, borrowing strength across geographic locations and incorporating prior information to obtain a posterior distribution for case trajectories. The compartmental model uses this distribution and predicts deaths using a random forest algorithm trained on COVID-19 data and population-level characteristics, yielding daily projections and interval estimates for cases and deaths in U.S. states. We evaluated the model by training it on progressively longer periods of the pandemic and computing its predictive accuracy over 21-day forecasts. The substantial variation in predicted trajectories and associated uncertainty between states is illustrated by comparing three unique locations: New York, Colorado, and West Virginia. The sophistication and accuracy of this COVID-19 model offer reliable predictions and uncertainty estimates for the current trajectory of the pandemic in the U.S. and provide a platform for future predictions as shifting political and societal responses alter its course.

## Author summary

COVID-19 models can be roughly classified as mathematical models that simulate disease within a population, including epidemiological compartmental models, or statistical curve-fitting models that fit a function to observed data and extrapolate forward into the future. Bridging this divide, we combine the strengths of curve-fitting statistical models

from https://github.com/COVID19Tracking/covid-tracking-data.

**Funding:** GLW, JAZ, PS, TB, MAS, and CMR received financial support from Private Health Management (https://www.privatehealth.com/) for this study. MAS was also supported through National Institutes of Health grant AI135995. The funders had no role in study design, data collection and analysis, decision to publish, or preparation of the manuscript.

**Competing interests:** I have read the journal's policy and the authors of this manuscript have the following competing interests: GLW, JAZ, PS, TB, and report personal fees from Private Health Management during the conduct of the study. CMR reports grants and personal fees from Private Health Management. MAS reports grants from US National Institutes of Health, grants from IQVIA, personal fees from Janssen Research and Development, and personal fees from Private Health Management during the conduct of the study. DX, LZ, JS, and AWR declare that they have no known competing financial interests or personal relationships that could have appeared to influence the work reported in this paper.

and the structure of epidemiological models, by embedding a Bayesian velocity model and a machine learning algorithm (random forest) into the framework of a compartmental model. Fusing these models together exploits the particular strengths of each to glean as much information as possible from the currently available data. We identify the velocity of log cumulative cases as an excellent target for modeling and extrapolating COVID-19 case trajectories. We empirically evaluate the predictive performance of the model and provide predicted trajectories with credible intervals for cumulative confirmed case count, active confirmed infections and COVID-19 deaths for each of the 50 U.S. states. Combining sophisticated data analytic methods with proven epidemiological models offers an empirically grounded strategy for making realistic predictions and quantifying their uncertainty. These predictions indicate substantial variation in the COVID-19 trajectories of U.S. states.

## Introduction

Rapid spread of SARS-CoV-2 virus across the planet has precipitated a global pandemic, killing millions and infecting tens of millions. Governments around the world have undertaken unprecedented interventions aimed at curtailing the spread and lethality of the virus. These interventions have relied heavily on predictions of COVID-19 case growth and mortality.

COVID-19 prediction models can be roughly classified as mathematical models that simulate disease within a population or statistical models that fit a function to observed data and extrapolate forward into the future. We will discuss the features of both types of models. Most COVID-19 models are compartmental models [1–61], a type of mathematical model used by epidemiologists to simulate infectious disease epidemics for over a century. Compartmental models divide a population into mutually exclusive compartments that denote disease status and supply a set of differential equations that define the flow of the population between compartments [62]. Traditionally they are named after their compartments with the SIR (susceptible-infectious-recovered) [63] and SEIR (susceptible-exposed-infectious-recovered) models as classic examples.

In an infectious disease compartmental model, $S(t)$ is the number of susceptible individuals at time $t$, and new infections are represented by the flow of individuals out of the S compartment. This is governed by the first derivative of $S(t)$ with respect to time, $dS(t)/dt$. Classic SIR and SEIR models express this as proportional to the product of $S(t)$, $I(t)$, and a rate constant $\beta$,

$$\frac{dS(t)}{dt} = \beta S(t)I(t), \tag{1}$$

where $I(t)$ is the number of infectious individuals at time $t$. The rate $\beta$ is often interpreted as disease transmissibility and may be expressed as a function of the reproductive number $R_0$—the expected number of individuals infected by an infectious person—and contact rates between individuals. It may also be normalized in Eq 1 by division by the total population size.

The simplest approach for simulating infections is to assume a value for $\beta$ or its constituent parts from the literature or other prior information [1–17]. While this is convenient, the predictive accuracy can suffer. Another approach that has been used by other studies is to estimate $\beta$ (or a related quantity) by fitting a statistical model or other optimization procedure to observed data [18–39]. This empirical approach can make these models more realistic, but they still may be limited in their ability to accurately model the COVID-19 pandemic. Disease transmission rates in COVID-19 have changed substantially over time depending upon the

political and societal responses and possibly other factors [54]. As a result, modelers operating within this framework often resort to modeling transmission rate changes by applying an adjustment factor that modifies transmission rates upward or downward in a somewhat ad hoc manner.

This has motivated modeling efforts that allow the disease transmission rate to vary over time, i.e., replacing $\beta$ in Eq 1 with $\beta(t)$ [40–50]. This is a promising approach, but to be useful for forecasting, estimates of $\beta(t)$ must extrapolate beyond the observed data to describe transmission at unobserved times and not simply interpolate the observed data, which is straightforward with a flexible model. Several studies have paired machine learning algorithms with COVID-19 compartmental models to accommodate time-varying effects, which may be useful at least when inference on the inputs to $\beta$ is not required. Yang et al. fit a long short-term memory neural network to data from the 2003 SARS outbreak adjusted by the output of their SEIR model [45]. Dandekar and Barbastathis augmented their compartmental models with a neural network that models time-varying transmission by estimating intervention efficiency from reported data as a function of time [42].

Recovery, death, and other states (e.g., hospitalization) may be incorporated into the model as separate compartments. Solutions to the differential system provide values for each compartment at each time, allowing for easy joint modelling of disease states once their derivative is specified. This is an advantage of compartmental models over many other approaches, which may require separate models for each quantity.

A number of agent-based COVID-19 models have been developed or adapted from influenza pandemic models to simulate the individuals of a population and their interactions [64–68]. This provides a mechanism for modelling interventions that target contacts between individuals and does not assume the population exists in homogeneous compartments as compartmental models generally do, but also requires a number of assumptions regarding the behavior and interactions within a population as well as the infectivity of COVID-19.

Serial growth models for COVID-19 simulate an epidemic by expressing the number of new infections at a given time as a weighted sum of new infections on previous days usually scaled by the reproductive number, which may be time-varying [69–72]. The weights are sampled from a probability distribution defining the amount of time between an individual being infected and infecting another person. Deaths or other outcomes may be modeled as a second step.

Statistical models often eschew deterministic population dynamics and fit the observed data as a function of time and possibly other covariates in a regression (or equivalent) framework. Log-linear [73], generalized Richards [74], ARIMA [75, 76], exponential [77], Gaussian CDF [78], and logistic [79–81] models, which all accommodate the generally sigmoidal shape of the cumulative infection count that is often observed in epidemics, as well as various other models [82–85] including machine learning algorithms [86–88] have been proposed for COVID-19. Murray et al. and Woody et al. take similar approaches for modeling COVID-19 deaths using the error function (ERF) [89, 90]. Count models (e.g., negative binomial) for the number of daily deaths is an alternative for modeling COVID-19 deaths [91]. Modeling deaths is appealing, because they have been more reliably reported than infections. However, because deaths lag infections by some amount of time, it may not enable projections to incorporate the latest information on disease spread.

Within the framework of a statistical (or other regression-like) model, it is easier to fit observed data, assuming an appropriate functional form is selected, but it may be challenging to accurately project the future trajectory of an epidemic. Time-varying covariates like mobile phone tracking data [90], Google trends [88, 92], and social media [93] are easily incorporated into such a model and may be quite predictive of the observed data. These data are not a

panacea, however, as forecasting requires knowledge of their values at future times, which are as yet unobserved. The forecasting accuracy of a model incorporating these covariates can depend heavily upon the accuracy of the assumptions made regarding their future values. Because of the challenges in jointly modeling multiple, non-Gaussian outcomes in a statistical model, regression approaches generally only model one outcome (e.g., infections or deaths) and additional steps must be taken to predict other quantities.

Here we project COVID-19 cases and deaths using a combination of Bayesian and machine learning data analytic methods to learn transition functions for a compartmental model. We introduce the velocity of log cumulative cases as a useful target for predicting case growth, and propose a Bayesian time series model that provides location-specific trajectories that extrapolate well within a full probability model, including uncertainty quantification. We use a random forest algorithm for the death transition function that learns the relationship between COVID-19 cases and population characteristics to predict deaths. We fuse the case and death models together by embedding them within a compartmental model that also provides projections for active cases and confirmed recoveries. The next section opens by introducing the data and presenting an overview of the model. Then it lays out in detail the Bayesian velocity model, the random forest death model, and the SIRD compartmental model. Lastly the paper closes with results and a discussion.

## Materials and methods

### Data

Daily COVID-19 confirmed cases and deaths for each state were obtained from the COVID Tracking Project, which combines information from state health departments and other sources [94]. The relationship between confirmed COVID-19 cases and the true number of infections is complicated, especially for the U.S., due to the substantial proportion of infections which are asymptomatic [95] and severely limited testing early in the pandemic [96]. Not only are confirmed cases a subset of COVID-19 infections, but the proportion of confirmed infections has differed across states and over the course of the pandemic as the prevalence and severity of cases as well as the availability of testing have changed. These difficulties pose challenges for basing a COVID-19 model on confirmed cases. As noted above, some modelers have focused on modeling deaths, since the death data is more reliable, and estimate infections in the preceding weeks as a second step [89, 91].

We model COVID-19 confirmed cases despite these challenges, because they are the best source of information on the current state of infections. The death data may be more accurate, but since deaths lag infections by several weeks they do not provide up-to-date insight into infections. While confirmed cases are a poor estimate of the total number of infections, they are still indicative of the prevalence and severity of disease spread. The shifting meaning of a confirmed case is indeed suboptimal, which motivates the use of a death model with a flexible mean structure that can learn the changing relationship between cases and deaths over the course of the pandemic.

### Model overview

There are three primary components to our model: (1) the velocity model for predicting new confirmed cases, (2) the death model for predicting how many cases end in death, and (3) a four compartment epidemiological model that fuses these together to provide joint predictions of cases, deaths and recoveries. The case model and the death model become transition functions within the compartmental model. There are several advantages to this combined approach. First, the SIRD model provides a joint model for cases, deaths and recoveries,

allowing simultaneous forecasting of these. This is an advantage over univariate models, including statistical regression models and machine learning prediction tools, which can only forecast one outcome. Second, the combined approach incorporates information on projected case growth into death predictions in a very flexible manner, which would not be available if the models were separate or if a less flexible death model were used. Third, the velocity model for projecting case growth both fits the observed data and extrapolates well, which is a challenge for curve-fitting approaches. Fourth, we incorporate uncertainty of model fit into the compartmental forecasts, by running it many times—once for each posterior sample from the Bayesian case model. R code for fitting the models, producing forecasting and generating figures is available at https://github.com/gregorywatson/covidStateSird.

## Bayesian velocity model for forecasting cases

We forecast new COVID-19 cases by modeling the velocity of the log cumulative cases. Forecasting COVID-19 cases in this velocity domain is appealing, because it reveals seemingly subtle shifts in case trajectory that are not obvious when considering raw case counts. Let $u_i(t)$ denote the cumulative case count for location $i$ at time $t$. The velocity (the first derivative with respect to time) of the log transformed cumulative cases is the instantaneous rate of new cases to cumulative cases at a given time,

$$\frac{d}{dt} \log u_i(t) = \frac{du_i(t)}{dt} \cdot \frac{1}{u_i(t)},$$

which is related to the reproductive number, but is readily estimated from the data. Calculating the reproductive number at a particular time, on the other hand, requires knowing the number of active infections. There is currently no reliable data on this, as most infections resolve on their own outside of a clinical or otherwise supervised setting in which their transition from active case to recovered might be recorded.

A crude estimate of the derivative can be obtained using first differences, but smoothing allows for more precise estimates, as calculating the derivative requires some notion of function smoothness [97]. We estimate the velocity by fitting a cubic spline to the observed log cumulative case count and then evaluating its derivative at the observed time points. Since there is relatively little noise in the cumulative counts, we assume any uncertainty introduced by this procedure is negligible.

Fig 1 depicts cumulative cases, log cumulative cases, and the velocity of log cumulative cases for 3 example U.S. states, New York (NY), Colorado (CO), and West Virginia (WV). The horizontal axis enumerates days since 100 or more confirmed cases were reported in that state, a milestone that proxies for the establishment of community transmission. Community transmission or its proxy is a sensible time point for data alignment, because there is substantial variation observed in the length of time between the detection of the first cases in a location and the acceleration of cases accompanying community transmission. This variation likely reflects both the possibility of containing a small number of initial cases and the increased uncertainty accompanying small samples.

The velocity of a cumulative function cannot be negative, since cumulative functions are monotonically increasing. Consequently, we employed a log link to map velocity to the entire real line and modeled it with a Bayesian autoregressive (AR-1) time series model. We estimated location-specific parameters, borrowing strength across locations for more precise estimates while accommodating individual variation. Borrowing strength can be particularly helpful for estimating the trajectory of locations with smaller populations or less advanced outbreaks.

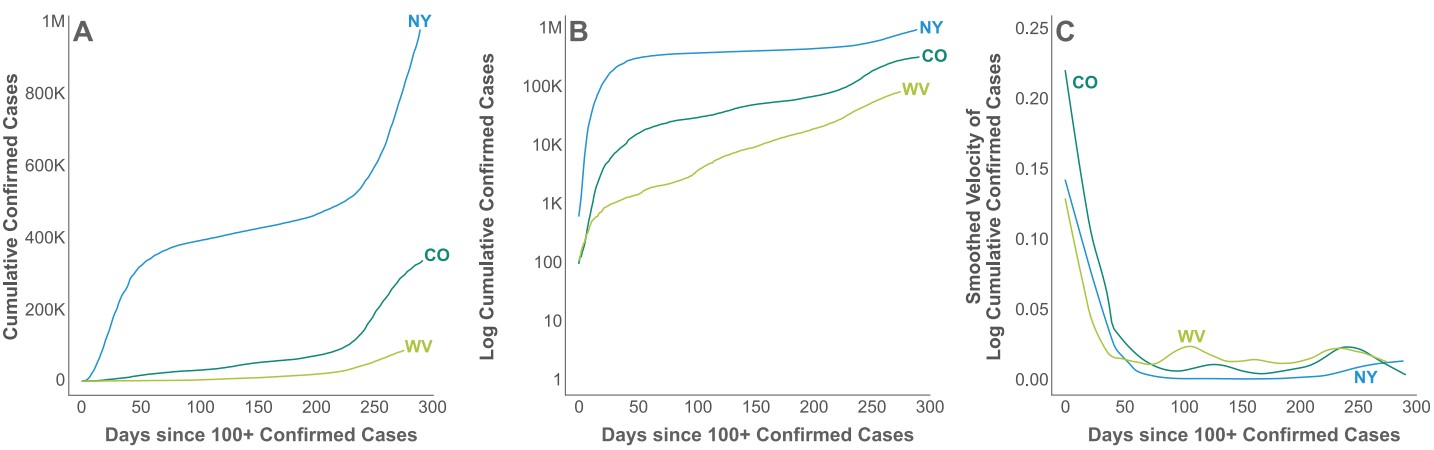

**Fig 1. Log cumulative cases and its velocity.** The cumulative case count (a), the log cumulative confirmed case count (b) and its velocity (c), i.e., first derivative with respect to time, for three example states, New York (NY), Colorado (CO), and West Virginia (WV) since 100 or more confirmed cases were reported.

Let $y_i(t)$ denote the derivative of log cumulative cases for location $i$ at time $t$, i.e., $y_i(t) = d \log u_i(t)/dt$. Log-transformed velocity is modeled as

$$\log y_i(t) = \mu_i + \phi_i \log y_i(t-1) + \epsilon_i(t), \tag{2}$$

where $y_i(t)$ is the velocity at time $t \in \{2, \ldots, n_i\}$, $\mu_i$ is a location-specific constant, $\phi_i$ a location-specific coefficient encoding the dependence on the previous time point, and $\epsilon_i(t)$ is independently distributed Gaussian noise with mean 0 and precision (inverse variance) $\tau_i$, i.e.,

$$\epsilon_i \mid \tau_i \sim N_{n_i}(\mathbf{0}, \tau_i^{-1} I_{n_i}), \tag{3}$$

where $N_{n_i}$ is an $n_i$ dimensional Gaussian distribution, its mean $\mathbf{0}$ is a vector of zeros, and $I_{n_i}$ is the identity matrix. The precision parameters were assigned a gamma prior distribution with mean $\mu_\tau$ and variance $\sigma_\tau^2$,

$$\tau_i \mid \mu_\tau, \sigma_\tau^2 \sim Gamma(\mu_\tau^2/\sigma_\tau^2, \mu_\tau/\sigma_\tau^2), \tag{4}$$

with $\mu_\tau$ itself having a gamma hyperprior. The location-specific constant $\mu_i$ is assumed to be nonpositive, since we know that the velocity must eventually go to zero. Consequently, it was assigned a negative lognormal prior distribution, i.e.,

$$-\log \mu_i \mid \mu_\mu, \sigma_\mu^2 \sim N(\mu_\mu, \sigma_\mu^2), \tag{5}$$

with $\mu_\mu$ having a Gaussian prior. The autoregressive coefficients $\phi_i$ were given a beta prior with mean $\mu_\phi$ and variance $\sigma_\phi^2$,

$$\phi_i \mid \mu_\phi, \sigma_\phi^2 \sim Beta\left(\left[\frac{1-\mu_\phi}{\sigma_\phi^2} - \frac{1}{\mu_\phi}\right]\mu_\phi^2, \left[\frac{1-\mu_\phi}{\sigma_\phi^2} - \frac{1}{\mu_\phi}\right]\mu_\phi^2\left[\frac{1}{\mu_\phi} - 1\right]\right), \tag{6}$$

with $\mu_\phi$ having a uniform hyperprior between 0 and 1. The prior mean and variance values used for the predictions presented here are listed in S1 Table.

Posterior inference was conducted via Markov chain Monte Carlo (MCMC) simulation using JAGS 4.3.0 and the `R2jags` [98] package of R. Three chains of 200,000 iterations each were run after a burn in of 10,000 iterations and thinned to save every 1,500th sample.

The posterior samples of this velocity model provide forecasts for $\log d \log u_i(t)/dt$, which we convert into a transition function for our compartmental model. Transition out of the

susceptible compartment is governed by an expression for $dS_i(t)/dt$. The number of individuals who are no longer susceptible is the number of cumulative cases, i.e., $u_i(t) = N_i − S_i(t)$, where $N_i$ is the total population of location $i$. Since $dS_i(t)/dt = −du_i(t)/dt$, we can convert our posterior distribution for $d \log u_i(t)/dt$ into a transition function. The autoregressive model for $\log d \log u_i(t)/dt$ in Eq 2 can be converted into an expression for $du_i(t)/dt$ for use in the compartmental model,

$$\frac{d}{dt}u_i(t) = u_i(t)\left[\frac{\frac{d}{dt}u_i(t-1)}{u_i(t-1)}\right]^{\phi_i} \exp\left[\mu_i + \frac{1}{2\tau_i}\right]\frac{S_i(t)}{S_i(t_0)}.$$

The details of this derivation are in S1 Appendix.

Noting that $N_i − S_i(t)$ gives the cumulative number of cases at time $t$ in the compartmental model described above, we set

$$dS_i(t)/dt = −du_i(t)/dt = −\xi_i(t).$$

The posterior mean or median of $−du_i(t)/dt$ could be used to estimate $\xi(t)$, but simply plugging in this single function into the SIRD model would ignore the uncertainty of this estimate. To incorporate this uncertainty explicitly into the SIRD model, we run the model separately for each posterior sample, giving a distribution of rate transition functions, $\xi_i(t)^{(1)}, \ldots, \xi_i(t)^{(m)}$. Accounting for uncertainty is important for COVID-19 forecasts, because without interval estimates quantifying uncertainty decision makers may place undue confidence in their accuracy.

## Death model

We constructed a random forest to predict deaths in each state on each day, using demographic characteristics of the state population and the number of COVID-19 cases and deaths reported on each of the preceding 21 days. This model would be useless for predicting deaths in most context, because lagged cases and deaths are unknown at future dates. However, within the compartmental model, it uses the case forecast provided by the velocity model in the previous section.

Random forest is a widely used heuristic machine learning prediction algorithm known to perform well at a variety of predictive tasks [99] by combining a large number of regression or classification trees into an ensemble [100]. We selected random forest for the death model over alternatives such as time series models, for 4 reasons: (1) in this context, we care only about predicting deaths given recent cases, deaths and other covariates rendering the interpretive and inferential advantages of time series models moot; (2) the flexible mean structure of random forest accommodates nonlinear effects, interactions and provides implicit variable selection, all of which are much more challenging in a time series context; (3) each death model prediction is only one day into the future, not an entire time series; and (4) the relationship between cases and deaths appears to have shifted in the U.S. throughout the course of the pandemic so far (for reasons that are not entirely clear—increased testing, better treatment protocols, a younger infected population, and viral attenuation may be contributing factors), suggesting that a nonstationary time series model would be needed, making the process of fitting such a model even more challenging.

Let $d_{ij}$ denote the number of dead reported in location $i$ on day $j$, where days are indexed for each location from the first day on which 100 or more cumulative confirmed cases were reported in that location. Let $\mathbf{w}_{ij} = (w_{ij1}, \ldots, w_{ijp})'$ denote the vector of $p$ covariates for location $i$ on day $j$. The conditional expectation of $d_{ij}$ given covariates $\mathbf{w}_{ij}$ is modeled as a random forest,

i.e., as an ensemble of bootstrapped regression trees,

$$\mathrm{E}d_{ij} \mid \mathbf{w}_{ij} = f(\mathbf{w}_{ij}) = \frac{1}{B} \sum_{b=1}^{B} T_b(\mathbf{w}_{ij}, \boldsymbol{\varphi}_b),\tag{7}$$

where $b = 1, \ldots, B$ indexes bootstrap samples of the training data, and $T_b(\mathbf{w}_{ij}, \boldsymbol{\varphi}_b)$ is a regression tree trained on the $b$-th bootstrap sample that relates covariate vector $\mathbf{w}_{ij}$ to parameters $\boldsymbol{\varphi}_b$. The model was fit using the `randomForest` package [101] of R using the default parameter values for the number of trees (500) and the number of covariates considered for each recursive split of the covariate space (floor($p/3$)). To quantify the uncertainty associated with random forest predictions, we follow the procedure devised by Zhang et al. to produce 95% prediction intervals from the out-of-bag errors [102, 103]. This results in a prediction interval for each run of the compartmental model. We take the fifth quantile of the lower bounds and the 95th quantile of the upper bounds to produce an overall prediction interval.

Fig 2 lists the covariates included in the model and their importance scores. Age, sex and comorbidity have been consistently reported in the literature as important risk factors for COVID-19 mortality. Even in the U.S. where testing has been limited, we expected that COVID-19 deaths on a particular day would be highly related to the number of cases and deaths reported on preceding days. Consequently the number of newly reported COVID-19 cases and deaths in location $i$ on days $t - 1, \ldots, t - 21$ were included as covariates for predicting deaths on day $t$.

Covariate importance scores were computed using permutation variable importance. Briefly, the permutation importance of a covariate is the decrease in predictive accuracy (in terms of mean squared error (MSE)) comparing the original model and a model in which that variable is randomly permuted to obscure any signal it might have with the outcome variable.

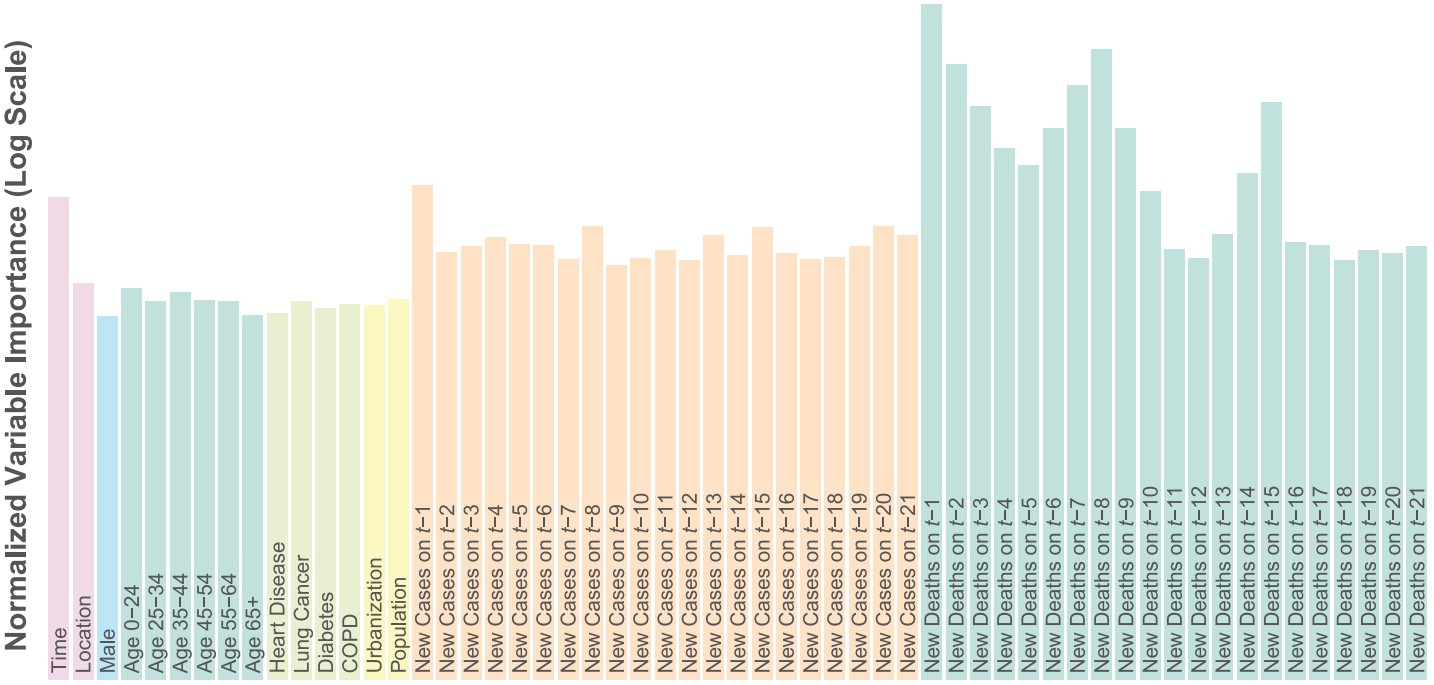

**Fig 2. Death model covariate importance.** Covariate importance scores on the log scale for the random forest death model as the mean decrease in MSE associated with permutation of the variable's values.

If a covariate is important in terms of prediction, obscuring its signal should result in a decrease in predictive accuracy. Not surprisingly, lagged deaths are highly important and to a lesser extent cases and time. Interestingly there appears to be a weekly periodicity to the lagged importance as there are peaks at $t - 1$, $t - 8$ and $t - 15$ especially for deaths. This is likely due to the effect of the workweek on data reporting. Additional lagged data beyond 21 days did not improve predictive performance, and so were not included in the model.

Fitting the model to data collected through December 31, 2020, resulted in an out-of-bag $R^2$ of 0.96. This is an overly optimistic estimate of prediction error, due to the within-location and temporal dependence of the data [104], but more significantly due to the lagged data being very informative covariates. Lagged deaths and cases were far more important than the demographic characteristics, which is not surprising considering the very strong relationship between testing positive for COVID-19 and dying of COVID-19, especially in the early days of the pandemic in the U.S., when testing was quite limited. Within the compartmental model, the lagged data are estimated, not observed, and so introduce uncertainty into the forecast. The random forest predictions were capped at a percentage of the new cases to avoid unrealistically high death predictions, which can occur when there are relatively few new cases. This upper bound was set to be equivalent to a 15% case fatality rate for the first 30 days of the epidemic and reduced to 7% subsequently, with the higher initial death rate motivated by the relative severity of early confirmed cases due to limited testing.

## The SIRD compartmental model

We combine the case and death models to forecast the spread and progression of COVID-19 through the populations of U.S. states using a SIRD compartmental model named after the four compartments into which it partitions the population: $S$ for susceptible, $I$ for infectious, $R$ for recovered, and $D$ for dead. The compartmental model allows for the joint forecasting of these quantities, a distinct advantage over many approaches including so-called black box prediction tools that generally only model a single outcome. The posterior samples from the velocity model provide a mechanism for uncertainty quantification that can be propagated through the compartmental model. The compartmental model also allows the case forecast to be used as covariates in the death model, which otherwise would not provide predictions beyond one day past the observed data.

The number of population members in each compartment is a function of time, $t$, and these functions are linked by a system of ordinary differential equations (ODEs) that govern the flow of the population through the different disease states:

$$
\begin{aligned}
\frac{dS(t)}{dt} &= -\xi(t), \\
\frac{dI(t)}{dt} &= \xi(t) - \rho I(t), \\
\frac{dR(t)}{dt} &= \rho I(t) - \theta(t), \\
\frac{dD(t)}{dt} &= \theta(t).
\end{aligned}
\tag{8}
$$

Fig 3 graphically depicts the SIRD model with arrows between compartments indicating possible transitions between compartments. Only deaths due to COVID-19 are permitted within this framework under the assumption that ignoring other causes of death, as well as the influx

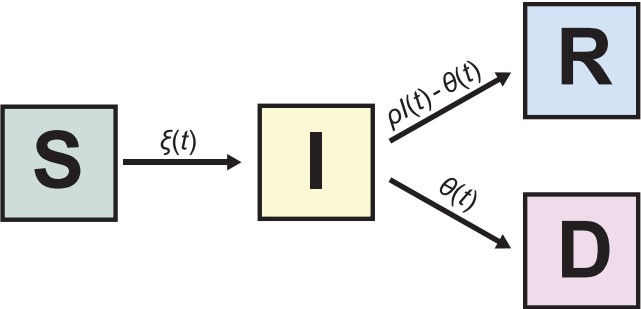

**Fig 3. The SIRD model.** Each of the four compartments quantifies the number of population members with that disease status: S for susceptible, I for infectious, R for recovered and D for dead. The arrows indicate possible transitions between disease states.

of new susceptible persons through birth or immigration, will not substantially alter inference in the short term.

The transition rates between compartments are determined by the functional forms and parameter values in Eq 8. Given these and initial conditions for the system, $S(t_0)$, $I(t_0)$, $R(t_0)$, and $D(t_0)$, the system of ODEs in Eq 8 is deterministic, but in general does not accommodate analytical solutions. Consequently, we compute numerical solutions using the `lsoda` solver in the `deSolve` package [105] of R [106].

Due to the novelty of the SARS-CoV-2 virus and a desire to empirically ground the compartmental model, we fit transition functions that can vary in time and incorporate covariates and other information. The transition between S and I is determined by $\xi(t)$, which describes the number of individuals becoming confirmed COVID-19 cases. This differs from traditional compartmental models. The standard expression for $dI(t)/dt$ is $\beta S(t)I(t)$ (sometimes divided by the population size $N$) as described in the introduction. We found the traditional functional form for $dI(t)/dt$ fit the observed data very poorly, which motivated its replacement by $\xi(t)$, a time-varying function derived from the velocity model described above. Importantly, $\xi(t)$ does not depend on $I(t)$, which is a departure from traditional compartmental models and is similar to the approach of so-called curve-fitting models. This hybrid approach was motivated by a desire to retain the benefits of compartmental models while exploiting the substantially better empirical accuracy of curve-fitting models for the changing number of cases.

Traditional SIR compartmental models use a rate parameter, which we call $\rho$, that is the inverse of the time an individual is expected to be infectious to model the movement of individuals out of the infectious compartment. We follow this approach, but split the R compartment into R and D, because we have reliable data on COVID-19 deaths, but not on recoveries. (Some states have reported recoveries, but in most instances this is limited to hospitalized patients who have recovered.) Like a traditional SIR model, we let $\rho I(t)$ denote individuals exiting the infectious compartment, which corresponds to the $-\rho I(t)$ term in $dI(t)/dt$. Since individuals do not enter compartment I until they test positive, in our model $\rho^{-1}$ is the length of time we expect an individual to remain infectious after testing positive. Using onset of symptoms as a proxy for testing positive, we sample $\rho^{-1}$ independently for each run from a Gaussian distribution with mean 10 and standard deviation 1, based on Wölfel et al. estimating the probability of isolating virus dropping below 5% at 9.78 days after symptom onset [107]. The death model, $\theta(t)$, indicates how many of these die, i.e., $dD(t)/dt = \theta(t)$, with the remainder of the $\rho I(t)$ recovering, i.e., $dR(t)/dt = \rho I(t) - \theta(t)$.

In addition to the SIRD forecasts of infections, deaths and recoveries, we estimate the effective reproductive number, $R_t$. This is the average number of new cases that each case will generate. We estimate this as

$$R(t) = \rho \frac{\xi(t)}{I(t)},$$

and report its ten-day moving average. We also include state-specific, time-varying estimates of case doubling time, death doubling time and the proportion of cases resolving in subject death.

A unique initial condition was constructed for each run of the compartmental model by stepping the model through each day of the observed data and fixing the number of cases and deaths to the observed values while using the recovery transition function to distribute cases between compartments I and R. This combines the observed case data while attempting to account for the uncertainty in the number of individuals in I and R using the randomness in the recovery function. Using the observed case data and incorporating uncertainty reduces the sensitivity of the model to the choice of initial conditions. This approach ignores any measurement error in the case and death data, which is a substantial limitation considering the status of COVID-19 case data in the US, as discussed above.

## Predictive accuracy

We assessed the predictive accuracy by training the model on case and death data collected through the end of August, September, October and November 2020, and forecasting the subsequent 21 days. We quantified prediction error for each state on each day using the mean absolute scaled error (MASE) of the posterior median number of new cases and deaths. MASE is computed by dividing the mean absolute prediction error by the in-sample mean absolute error (MAE) of a naive random walk forecast,

$$MASE(\mathbf{Y}, \mathbf{Y}^*, \hat{\mathbf{Y}}) = \frac{\frac{1}{m}\sum_{j=1}^{m}|Y_j^* - \hat{Y}_j|}{\frac{1}{n-1}\sum_{i=2}^{n}|Y_i - Y_{i-1}|}, \tag{9}$$

where $\mathbf{Y} = (Y_1, \ldots, Y_n)'$ is the training data outcome, $\mathbf{Y}^* = (Y_1^*, \ldots, Y_m^*)'$ is the observed outcome in the evaluation set and $\hat{\mathbf{Y}} = (\hat{Y}_1^*, \ldots, \hat{Y}_m^*)'$ is the prediction for $\mathbf{Y}^*$ to be evaluated [108]. MASE is scale invariant, which makes comparisons of predictive accuracy between states with epidemics on different scales more meaningful. A MASE of 1 indicates that the predictions were on average equally accurate to the mean accuracy of a random walk forecast in the training data. This is a somewhat conservative estimator of prediction error for COVID-19, because cases and deaths have generally increased with time, which means the MAE of a random walk forecast in the training data will be lower than the MAE of a random walk forecast in the subsequent evaluation data.

Fig 4 depicts the median and interquartile range of MASE across states for cases and deaths over a three-week forecast after each of the training periods. As expected, the median and interquartile range of the MASE increased for both cases and deaths as forecasts extrapolated farther from the training data, although this increase is only slight for deaths. The model predicted cases and deaths reasonably well in light of the conservativeness of the estimator, especially within the first week of extrapolation, with the median MASE mostly below 1. The model forecasts deaths over this period particularly well, with only slightly diminished accuracy at 21 days. This is due at least in part to the lagged relationship between cases and deaths,

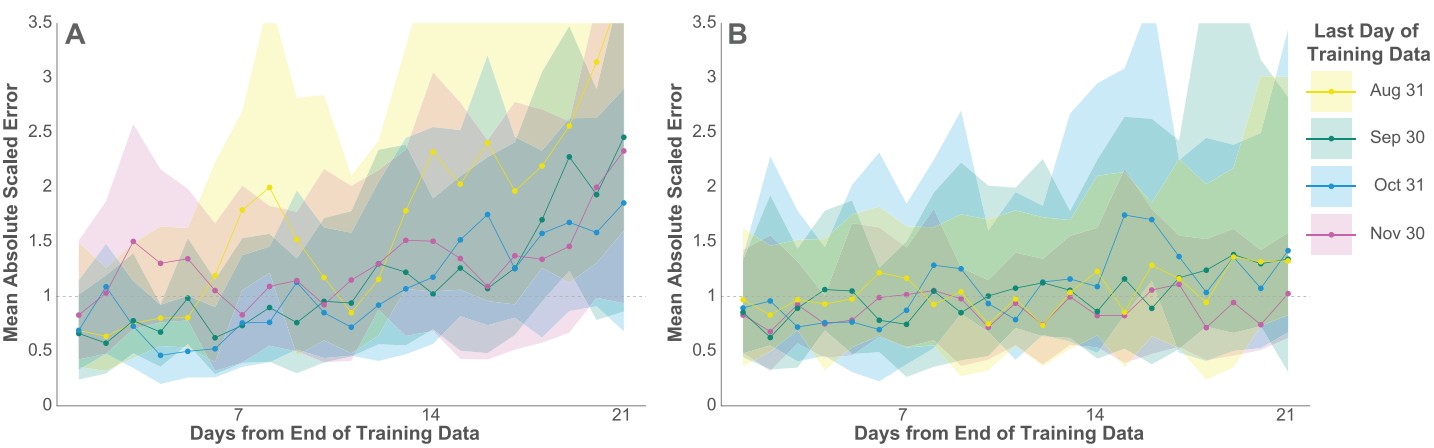

**Fig 4. Predictive accuracy.** The median and interquartile range (IQR) of MASE across all 50 states on each day of the 21-day prediction periods for new confirmed cases (a) and deaths (b). A MASE of 1 indicates equivalent accuracy to a one-day random walk forecast in the training data.

which makes case data much more informative for a 3-week death forecast than for a 3-week case forecast.

As an additional test of the death model's predictive accuracy, we compared it with a state-specific autoregressive (AR-1) model over the same 4 training and evaluation sets. The random forest death model predicted deaths more accurately than the AR-1 model for 3 of these 4 sets. The details of this evaluation may be found in S2 Table.

## Results & discussion

Infections and deaths were projected through April 1, 2021, for all 50 states. Fig 5 depicts median predicted cumulative confirmed cases as well as active confirmed infections and daily death counts for New York, Colorado, and West Virginia. These three states were selected as examples, because they are diverse in their population size, geography, political alignment, demographics, and in the progression of their COVID-19 epidemics. The equivalent figures for all 50 states are included in S2 Appendix.

New York, especially New York City with its large, dense population, was the epicenter of a large, early COVID-19 outbreak in the United States with over 300,000 confirmed cases by late April. Initial exponential case growth was slowly curbed by public interventions, leading to a consistent decrease in case velocity and peaks in active cases and deaths in mid April. Case growth being well past its peak translates into a plateaued cumulative case curve, which began to increase again in late 2020.

Colorado, in contrast, has had many fewer cases than New York with approximately 350,000 cases by the end of 2020. Rather than exhibiting a sharp peak followed by low case growth, Colorado cases exhibit a steady climb punctuated by waves of faster and slower growth. Its interval estimates are relatively wider than New York, because there is more uncertainty in the estimated trajectory. Colorado also exhibits more relative variation in its daily death counts than New York because of the smaller number.

West Virginia approaching 100,000 cases through the end of 2020 illustrates the estimated trajectories of a relatively rural state with slow case growth for the first few months of the pandemic, now showing signs of exponential growth. With cases growing more rapidly, there is correspondingly more relative uncertainty in its trajectory.

The figures include 95% credible intervals around the median indicating that 95% of simulation results fell within this region. These intervals are not true credible intervals in the

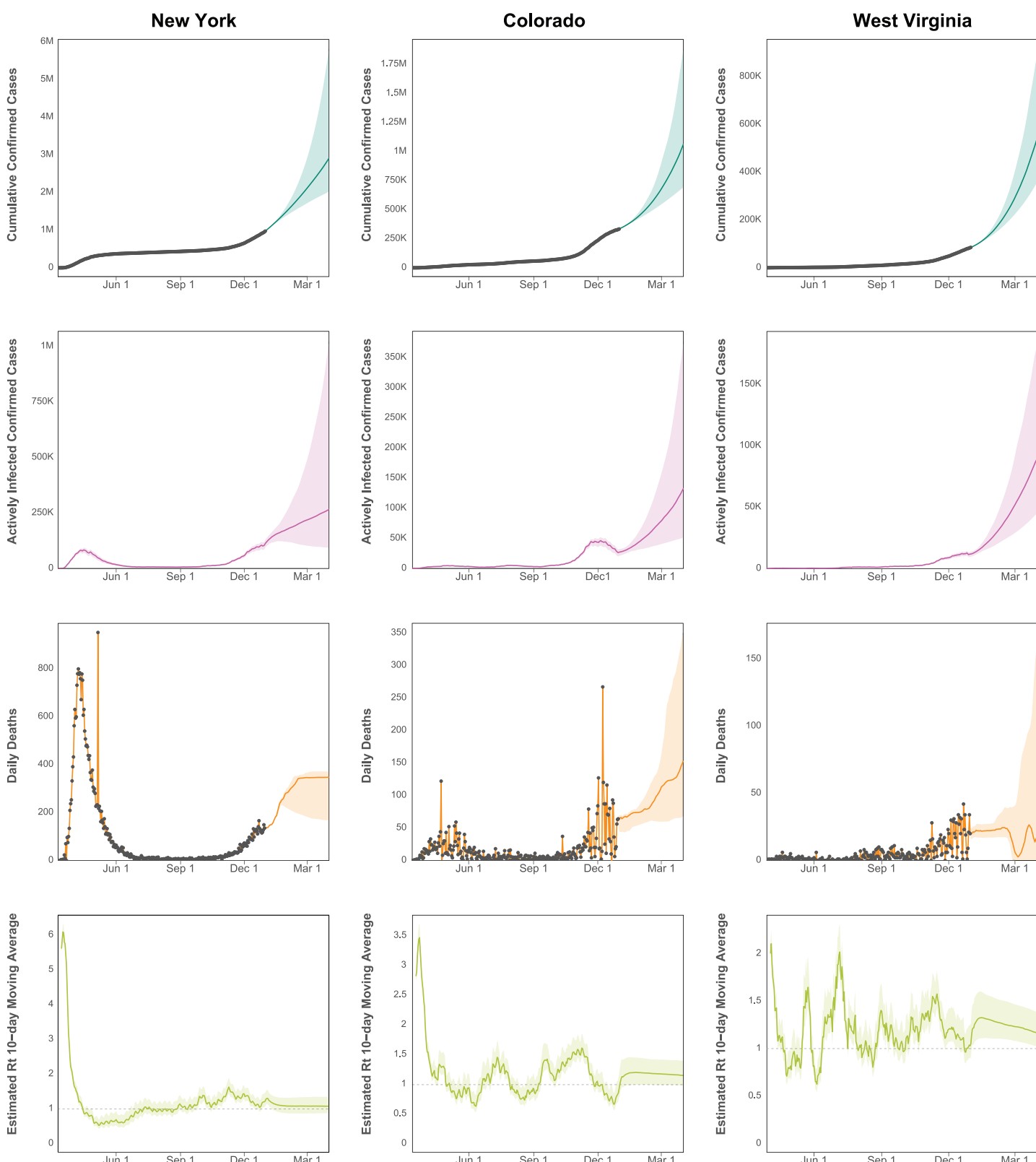

**Fig 5. Predicted cumulative cases, active infections, deaths, and effective $R_t$.** Projected cumulative case count, active confirmed infections, and daily deaths through April 1, 2021, for New York, Colorado, and West Virginia. The grey dots indicate observed data, which are not available for active infections and $R_t$.

Bayesian sense, because random forest is not a probability model. Nevertheless, they represent a reasonable account of model uncertainty, as they incorporate credible intervals from the Bayesian case model, uncertainty around the duration of illness, and interval estimates for the random forest predictions.

These forecasts extrapolate foward the trajectory of the pandemic at the end of 2020, but its future depends upon the ongoing societal and political response to the pandemic, and will be altered by future events. For example, aggressive lockdowns have been used to blunt case trajectories as in many other countries. Similarly vaccinations and newly developed treatments will substantially impact the number of new cases or deaths.

Despite the strengths of the current approach, it is not without limitations. The projections produced here assume states continue upon their current trajectories. Changes in policy interventions, for example, may result in substantial deviation from this. Projecting outcomes under different or changing intervention scenarios is the subject of ongoing work.

Considering COVID-19 cases and death over large areas can obscure variation on a smaller scale. It is possible for a generally positive trajectory at the state-level to mask a burgeoning outbreak in some locale within the state until that outbreak contributes sufficiently many cases to influence the state-wide trajectory. A more granular approach that models COVID-19 at a finer resolution may be able to identify such an outbreak earlier.

There is substantial interest in estimating the proportion of the population that has or will have recovered from COVID-19 in the hopes that these individuals have acquired at least temporary immunity to the virus and can be the vanguard to economic recovery. Since we focus on modeling confirmed cases and deaths, our model does not predict the true number of recovered individuals. It is well known that, especially in the U.S., confirmed cases are a substantial undercount for the true number of COVID-19 infections. As a result, estimating the number of recovered individuals requires additional information beyond predictions of confirmed cases and deaths. Attempts to quantify recovery using serology testing are underway in the U.S. and elsewhere.

Without the addition of covariates, the time series velocity model may not predict future case spikes, which may result from a return to pre-social distancing behavior or a change in governmental intervention. It does, however, accommodate these types of events quite well. The increasing velocity associated with a spike in cases corresponds to exponential growth at an increasing exponential rate. This rapidly causes an explosion of cases that pushes case growth beyond whatever level a particular population deems tolerable. In every case there has been a subsequent return to a velocity that corresponds to a tolerable level of case growth. By targeting this velocity, our model forecasts reasonable long-term case trajectories without needing to predict the occurrence of case spikes, which are quite difficult to anticipate precisely.

Finally, one could consider more elegant methods for incorporating lagged case and death counts into a death model than simply inserting them as covariates into random forest. However, many approaches to lag estimation are only good retrospectively and thus are insufficient for the current task.

This modeling framework suggests a number of avenues for future work. The most salient of these is the simulation of various scenarios that model policy or public health responses to the pandemic including the effects of vaccinations. Forecasting COVID-19 cases and deaths under alternate scenarios may provide useful information for decision makers. Future methodological improvements could include integrating all the components of the model within a single Bayesian model by substituting Bayesian additive regression trees (BART) for the random forest death model. This would provide a posterior distribution for all parameters and forecasts.

## Supporting information

**S1 Appendix. Derivation of case transition function.** The derivation of the compartmental model transition function from the autoregressive velocity model.
(PDF)

**S2 Appendix. State forecasts.** Projected cumulative case count, active confirmed infections, and daily deaths through April 1, 2021, for each of the 50 U.S. states.
(PDF)

**S1 Table. Parameter values & distributions.** The parameter values and prior distributions for the parameters of each component of the model.
(PDF)

**S2 Table. Death model comparison.** A comparison of the random forest death model to a state-specific autoregressive model over 4 different training and evaluation sets.
(PDF)

## Acknowledgments

We thank Donatello Telesca, Jay J. Xu, and Ian Frankenburg (University of California, Los Angeles) for their helpful comments and assistance.

## Author Contributions

**Conceptualization:** Gregory L. Watson, Anne W. Rimoin, Marc A. Suchard, Christina M. Ramirez.

**Data curation:** Di Xiong, Lu Zhang, John Shamshoian, Phillip Sundin, Teresa Bufford, Christina M. Ramirez.

**Formal analysis:** Gregory L. Watson.

**Funding acquisition:** Christina M. Ramirez.

**Investigation:** Gregory L. Watson, Di Xiong, Lu Zhang, John Shamshoian, Phillip Sundin, Teresa Bufford, Christina M. Ramirez.

**Methodology:** Gregory L. Watson, Marc A. Suchard, Christina M. Ramirez.

**Project administration:** Christina M. Ramirez.

**Resources:** Christina M. Ramirez.

**Software:** Gregory L. Watson, Di Xiong, Lu Zhang, Joseph A. Zoller, John Shamshoian, Phillip Sundin, Teresa Bufford, Marc A. Suchard.

**Supervision:** Christina M. Ramirez.

**Validation:** Gregory L. Watson, Christina M. Ramirez.

**Visualization:** Gregory L. Watson.

**Writing – original draft:** Gregory L. Watson.

**Writing – review & editing:** Marc A. Suchard, Christina M. Ramirez.

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
