## [Decision Letter · Decision Letter 0]

1 Jul 2020

Dear Mr. Watson,

Thank you very much for submitting your manuscript "Fusing a Bayesian case velocity model with random forest for predicting COVID-19 in the U.S." for consideration at PLOS Computational Biology.

As with all papers reviewed by the journal, your manuscript was reviewed by members of the editorial board and by several independent reviewers. In light of the reviews (below this email), we would like to invite the resubmission of a significantly-revised version that takes into account the reviewers' comments.

We cannot make any decision about publication until we have seen the revised manuscript and your response to the reviewers' comments. Your revised manuscript is also likely to be sent to reviewers for further evaluation.

Sincerely,

Benjamin Althouse

Associate Editor

PLOS Computational Biology

Virginia Pitzer

Deputy Editor

PLOS Computational Biology

Reviewer's Responses to Questions

**Comments to the Authors:**

Reviewer #1: The method proposed by Watson and colleagues has various compelling aspects--it combines a Bayesian hierarchical model, compartmental SIRD model and a machine learning methods for forecasting into the future--with the goal of accurately forecasting COVID-19-confirmed cases and deaths. The model is meant to predict so-called “case velocity” and yields forecasting intervals, which is very important, and I am not concerned by the fact that they cannot be formally interpreted as posterior uncertainty intervals. Overall, while I think that the approach taken is a sensible one, there are a number of concerns and suggestions that I would make to the authors towards a much stronger revised version of the paper.

1. Each of the modeling components of your method has its strengths and weaknesses. The strengths of the methods are not explained or utilized to the fullest degree possible; the weaknesses should be more fully explored through sensitivity analyses. I will taken each component in turn.  a. The strengths of the SIRD model are that it is mechanistic, however this is not exploited to, e.g. simulate various scenarios for opening up, etc. Estimates of the rate of transmission by state are also of particular importance—and can be compared to the estimates derived by other methods—but it does not seem that these are explicited reported anywhere unless I have missed them. A limitation of the SIRD model is that it requires setting initial conditions—your discussion on lines 120-123 should be expanded to understand the sensitivity our your approach to these choices. I am not an expert on SIRD, but I imagine as with any mechanistic model there are potentially many sensitivities to somewhat arbitrary choices. These should be explored.  b. The strengths of the hierarchical model are that it gives interpretable state-level estimates for parameters that policymakers care about. I saw little discussion of the posteriors over these parameters, how they compare to previous studies, etc. There are of course limitations here, which could be discussed as well, but I don’t expect this to drastically change your overall results in any way.  c. The strengths of the random forest model are that it is an effective black box machine learning method that can use whatever covariates you throw at it. The limitations are that it is only partially interpretable—variable importance is useful, but does not give effect sizes. It would be very useful to know if a simpler, interpretable method were able to give reasonable results. Two ideas that you may have considered and would be worth a quantitative comparison: time series methods (e.g. autoregressive or exponential smoothing); carrying forward the SIRD model

2. Your SIRD formulation is non-standard, as far as I am aware. I would have expected theta(t) to appear in the equation for dI/dt. Is your Eq (2) somehow equivalent to the following, which is what I expected?

dS/dt = -xi(t)

dI/dt = xi(t) - rho I (t) - theta(t)

dR/dt = rho I(t)

dD/dt = theta(t)

3. I am not totally convinced by how the three parts of your model fit together.

a) It would be nice if there was more integration between the pieces—there are methods for Bayesian inference of SIR models using modern probabilistic programming languages like Stan. Could you have combined the hierarchical model with the SIRD model? I will not claim this is easy as I have not tried, but it would be good to discuss this.

b) the discussion surrounding Eqs 8-9 says that c_i is not identifiable and you find it by minimizing MSE, but I am not sure what c_i is and would like to know how well this minimization works and what effect it has. I’m also not convinced that this yields a posterior distribution for c_i, but given that ultimately you are not doing Bayesian inference it probably doesn’t matter.

c) I am trying to make sense of sampling rho ~ Normal(14,1). Given your formulation of SIRD (which I questioned earlier), rho is the probability of transitioning from Infection to either Recovered or Death. You should provide citations to justify a mean of 14. I am familiar with estimates of the infection-to-onset distribution and the onset-to-death distribution, but this is neither of those.

d) The random forest component of your approach is meant to give an estimate of the transition from infection to death, but I am not convinced that this is what it is truly doing due to my concerns about the reliability of case reports (see below). To understand and diagnose this concern, it would be helpful to compare your random forest model’s predictions of the number of people transitioning from I to D to published estimates of the infection fatality rate. Admittedly, I’m a little unclear on whether/how these can be directly compared, so if this is impossible it would be useful to know why as I see this as a limitation of your model, despite it having an underlying epidemiological component. Returning to the infection-to-death distribution, estimates put this at around 2 to 3 weeks, so it seems like your random forest model should include data that is more than 14 days in the past. (In fact, I don’t see what is the relevance of data t-1 or t-2!) The data limitations you mention can now be alleviated as I imagine you will want to refit your model on the most recently available data.

4. The reliance on cases concerns me with respect to accurately modeling COVID-19 due to the widespread, undetected community transmission in the early period of the USA epidemic, the changing testing strategies and availability during the course of the USA epidemic, and the large fraction of asymptomatic cases throughout. Attempts have been made to adjust for some of this (e.g. Hsiang et al, Nature 2020) but I am not convinced that this is truly possible, which is why my group has focused on modeling deaths instead of cases (Flaxman et al, Nature 2020). This all leads to another important concern, which is that your overall goal is predict case velocity. I can see this as potentially a useful goal, especially in the post-stay-at-home phase of the epidemic and the possible second wave phase of the epidemic, but would like to see more justification for this goal. I am still not convinced that case reporting is good enough that we should rely on it—if it is not good enough, then why is it your goal to predict it? Said differently, perhaps you wish to predict the true rate of cases; but then what, exactly is a case? The true thing that you could try to predict is infections. I am certainly not saying you need to take the approach of our group, but for work in this vein see Flaxman et al Nature 2020 and our US report: https://www.imperial.ac.uk/media/imperial-college/medicine/mrc-gida/2020-05-28-COVID19-Report-23-version2.pdf

5. If the overall goal is accurate forecasting of case velocity, then I think there are two pieces missing:

a. A more indepth and exhaustive approach to evaluating your forecasting method—this could, e.g. include 1, 3, 7, and 21 day ahead forecasting, starting from an early period in the epidemic and carried forwards with subsequently more days of data included. Various metrics could be included, including MAE/coverage/CRPS.

b. A justification for including the SIRD and hierarchical modelling framework; would a black box machine learning method on its own give at least as good performance as your method? If so, what is it that your model reveals about the spread of the disease that a black box model on its own would not?  

6. Source code to replicate every part of your analysis should already be made available. In this world of falling public trust in scientists and policymakers, it is critically important to strive for replicable and reproducible analyses. The effort spent is entirely worth it due to the benefit that comes from having more pairs of eyes inspecting your assumptions, your model, your data pipeline, and so on.

7. Speaking of data, I may have missed it but do you use JHU or NYT data? We found that JHU data was on the whole reliable, though it suffers from various inconsistencies, and it is important to point out that this data is usually date of report for deaths, rather than date of death. (The JHU data was not reliable for New York State, and for this we used NYT.)

8. A minor issue, and I am sorry to quibble about this but it is of course the part I know best: I do not agree with your characterisation of the Flaxman et al model as a serial growth model (honestly I’m not sure what this means so maybe I’m wrong!), nor is the description of “weights being sampled from a probability distribution” an accurate description of our Bayesian inference method (perhaps it is an accurate description of the other methods). Compare, for example, our fully Bayesian MCMC scheme to the number of infections on a given day (which derives from a discrete time convolution using the generation distribution multiplied by the time-varying reproduction number R) to your scheme for introducing stochasticity into your SIRD model by sampling rho which could accurately be described as weights being sampled from a probability distribution.

Signed,

 Seth Flaxman

Reviewer #2: please see attached

**Have all data underlying the figures and results presented in the manuscript been provided?**

Reviewer #1: **No: **see comments about reproducibility in the review

Reviewer #2: Yes

PLOS authors have the option to publish the peer review history of their article (what does this mean?). If published, this will include your full peer review and any attached files.

Reviewer #1: **Yes: **Seth Flaxman

Reviewer #2: **Yes: **Spencer Woody
---

## [Decision Letter · Decision Letter 1]

12 Nov 2020

Dear Mr. Watson,

Thank you very much for submitting your manuscript "Fusing a Bayesian velocity model with random forest for predicting COVID-19 in the U.S." for consideration at PLOS Computational Biology. As with all papers reviewed by the journal, your manuscript was reviewed by members of the editorial board and by several independent reviewers. The reviewers appreciated the attention to an important topic. Based on the reviews, we are likely to accept this manuscript for publication, providing that you modify the manuscript according to the review recommendations.

Sincerely,

Benjamin Muir Althouse

Associate Editor

PLOS Computational Biology

Virginia Pitzer

Deputy Editor

PLOS Computational Biology

[LINK]

Reviewer's Responses to Questions

**Comments to the Authors:**

Reviewer #1: Dear authors,

Thank you for your comprehensive response and set of revisions. In particular, your clarification and focus on "case velocity" has definitely sharpened the paper and its goals.

There are a few final issues that I think you should address to strengthen your paper and its support for the new method you have proposed.

1) I still think it would be very worthwhile to address this point: "The random forest component of your approach is meant to give an estimate of the transition from infection to death, but I am not convinced that this is

what it is truly doing due to my concerns about the reliability of case reports (see below).To understand and diagnose this concern, it would be helpful to compare your random forest model’s predictions of the number of people transitioning from I to D to published estimates of the infection fatality rate." If helpful, a meta-analysis on IFR is now available https://www.imperial.ac.uk/mrc-global-infectious-disease-analysis/covid-19/report-34-IFR/

2) You write: "The parameters of the velocity model are not directly comparable to the transmission rate at the heart of more traditional compartmental models". This clarification makes sense to me, but I do still wonder if you could include some direct comparisons to help the reader get a handle on how velocity works, by comparing it to more widely used (and I would argue better understood) quantities such as: the reproduction number, growth rate, or doubling time. Perhaps, for example, you could forward simulate from your model and then empirically calculate the doubling time?

3) Your points beginning "We agree with this characterization of the strengths and limitations of random forest." make sense to me; thank you for the clarification. I would quibble with this part, however, "It may be possible to construct a time series model that rivals the predictive performance of random forest for our purposes, but we are doubtful, and devising such a model would be a challenging project in its own right and beyond the scope of the

current work." Since your focus is on predicting one day ahead, I really think it would be helpful for you to include a very simple baseline comparison, such as an autoregressive model, trained separately for each state, or even a no change / constant model. If your model is significantly better than this model, great; if it is competitive, that is fine as well, as you have persuasively argued that there is something to be gained by focusing on velocity rather than simply predicting case counts.

-Seth Flaxman

Reviewer #2: Please see attached comments.

**Have all data underlying the figures and results presented in the manuscript been provided?**

Reviewer #1: None

Reviewer #2: None

PLOS authors have the option to publish the peer review history of their article (what does this mean?). If published, this will include your full peer review and any attached files.

Reviewer #1: **Yes: **Seth Flaxman

Reviewer #2: **Yes: **Spencer Woody
---

## [Decision Letter · Decision Letter 2]

26 Feb 2021

Dear Mr. Watson,

We are pleased to inform you that your manuscript 'Pandemic velocity: forecasting COVID-19 in the US with a machine learning & Bayesian time series compartmental model' has been provisionally accepted for publication in PLOS Computational Biology.

Best regards,

Benjamin Muir Althouse

Associate Editor

PLOS Computational Biology

Virginia Pitzer

Deputy Editor-in-Chief

PLOS Computational Biology

Reviewer's Responses to Questions

**Comments to the Authors:**

Reviewer #1: I appreciate the work you have done to revise the paper and hope that it is published soon!

**Have all data underlying the figures and results presented in the manuscript been provided?**

Reviewer #1: Yes

PLOS authors have the option to publish the peer review history of their article (what does this mean?). If published, this will include your full peer review and any attached files.

Reviewer #1: **Yes: **Seth Flaxman

---

## [Editor Report · Acceptance letter]

25 Mar 2021

PCOMPBIOL-D-20-00822R2 

Pandemic velocity: forecasting COVID-19 in the US with a machine learning & Bayesian time series compartmental model

Dear Dr Watson,

I am pleased to inform you that your manuscript has been formally accepted for publication in PLOS Computational Biology. Your manuscript is now with our production department and you will be notified of the publication date in due course.

With kind regards,

Katalin Szabo
